# Impact of Public Policy and COVID-19 Pandemic on Hepatitis C Testing and Treatment in France, 2014–2021

**DOI:** 10.3390/v16050792

**Published:** 2024-05-16

**Authors:** Cécile Brouard, Manon Schwager, Aude Expert, Nicolas Drewniak, Stella Laporal, Grégoire de Lagasnerie, Florence Lot

**Affiliations:** 1Santé Publique France, The National Public Health Agency, 94415 Saint-Maurice, France; nicolas.drewniak@santepubliquefrance.fr (N.D.); stella.laporal@santepubliquefrance.fr (S.L.); florence.lot@santepubliquefrance.fr (F.L.); 2Caisse Nationale de l’Assurance Maladie, 75986 Paris, France; manon.schwager@assurance-maladie.fr (M.S.); aude.expert@assurance-maladie.fr (A.E.); gregoire.de-lagasnerie@assurance-maladie.fr (G.d.L.)

**Keywords:** hepatitis C virus, testing, treatment, direct-antiviral agent, COVID-19, French National Health Insurance Data System (SNDS), France

## Abstract

Given the World Health Organization’s target to eliminate the hepatitis C virus (HCV) by 2030, we assessed the impact of French public policies and the COVID-19 pandemic on HCV testing and initiation of direct-antiviral agents (DAAs). Using the French National Health Data System, we identified individuals living in metropolitan France with at least one reimbursement for an anti-HCV test and those with a first delivery of DAAs between 1 January 2014 and 31 December 2021. During this period, the annual number of people tested increased each year between 3.3 (in 2015) and 9.3% (in 2021), except in 2020, with a drop of 8.3%, particularly marked in April (−55.0% compared to February 2020). A return to pre-pandemic testing levels was observed in 2021. The quarterly number of patients initiating DAAs presented an upward trend from Q1-2014 until mid-2017, with greater increases in Q1-2015, and Q1- and Q2-2017, concomitant with DAA access policies and availability of new therapies. Then, quarterly numbers decreased. A 65.5% drop occurred in April compared to February 2020. The declining DAA initiations since mid-2017, despite new measures improving access and screening efforts, could be due to the shrinking pool of patients requiring treatment and a need to increase awareness among undiagnosed infected people. Further action is needed to eliminate HCV in France.

## 1. Introduction

In 2014, following the introduction of new direct-acting antivirals (DAAs), it became possible to eliminate hepatitis C virus (HCV) infection, which was set as the World Health Organization’s (WHO) elimination target by 2030 [1]. This therapy is highly effective (sustained virologic response over 95% for all genotypes), well tolerated, and pangenotypic, with a short treatment duration (currently 8–12 weeks for most patients) [2]. To achieve the WHO’s goal, it is thus necessary to monitor the progress of prevention strategies, particularly screening, diagnosis, and linkage to care.

In 2018, France committed to eliminating HCV by 2025, with a mid-term objective of 120,000 people treated with DAAs by late 2022 [3]. Indeed, HCV has been a public policy challenge in France for many years, with three national plans implemented since the 1990s [4]. Initially used in clinical trials and compassionate use programs, from December 2014 DAAs became available to patients with severe liver disease (fibrosis at least “severe” F2) or with comorbidities (i.e., HIV, cryoglobulinemia, and B-cell lymphoma), with an eligibility assessment conducted by a multidisciplinary team in HCV reference centers. Furthermore, DAAs could only be prescribed and delivered in hospital settings [5]. From July 2016, DAAs became available to patients with milder severe disease (F2, F3, and F4 fibrosis) and people at high transmission risk (e.g., incarcerated individuals, individuals who inject drugs, pregnant women) [6]. Finally, universal access was recommended by the French National Authority for Health (HAS) in December 2016 [7]. Other measures were implemented to enhance and facilitate access to DAAs, such as the end of eligibility assessments by multidisciplinary teams except for complex cases (August 2017), the extension of DAA dispensing to retail pharmacies (March 2018), and the authorization for all physicians to prescribe pangenotypic DAAs for patients without severe liver disease, previous DAA therapy, and comorbidities (May 2019) [8]. Concomitantly, several actions were implemented to encourage HCV testing, notably the authorization to perform rapid diagnosis tests in healthcare, social services, and community-based settings (2016) [9], the revision of the HCV testing strategy (2019), which concluded that screening should continue to target at-risk people [10], and the implementation of regional screening weeks annually since 2019 [11].

The most recent data on HCV testing and treatment in France concern the pre-COVID-19 period and show that 71,466 patients initiated HCV treatment between 2015 and 2019, while HCV screening increased between 2014 and 2019, which, according to Pol et al. [12], occurred due to the availability of DAAs.

Our main objective is to provide updated trends for the number of people tested for anti-HCV and the number of patients initiating DAAs for the period 2014–2021, and to describe their demographic characteristics in order to assess the impact of French public policies for HCV elimination on HCV testing and treatment. Our secondary objective is to study the effect of the COVID-19 pandemic on HCV screening and treatment initiation during the period 2020–2021.

## 2. Materials and Methods

### 2.1. Data Source

This observational study used comprehensive data from the French National Health Data System (SNDS), which contains individual anonymous data used for billing and reimbursement of all outpatient healthcare consumption, including prescribed drugs and laboratory tests, for individuals affiliated to the different health insurance schemes in France (99% of French residents, i.e., over 66 million inhabitants). The SNDS also includes the national hospital database, which contains some of the data relating to all public and private hospital stays. 

Sociodemographic data such as gender, age, and area of residence are documented for each individual [13]. However, due to the administrative purpose of the SNDS database, some epidemiological and clinical data are not available: risk factors, results of laboratory tests, histology or radiology, procedures or drugs not prescribed, not billed or not reimbursed (at an individual level), diagnosis of medical consultations, etc.

### 2.2. Study Populations and Study Periods

Two study populations living in metropolitan France were considered: individuals tested for anti-HCV and patients initiating DAAs.

To address the two objectives, two study periods were defined: (1) “DAA period” (1 January 2014–31 December 2021), to study the impact of public policy on HCV testing and DAA initiations; (2) “COVID period” (1 January 2020–31 December 2021), to assess the impact of the COVID-19 pandemic on HCV testing and treatment.

#### 2.2.1. Identification of Tested Individuals 

People tested for anti-HCV comprised all individuals with at least one reimbursement for an enzyme immunoassay test to detect anti-HCV antibodies (coded according to the Nomenclature of Medical Biology Acts) during the study period, whether performed in private hospitals, public hospitals (only outpatient tests), or ambulatory care settings. 

#### 2.2.2. Identification of Patients Initiating DAAs 

All people receiving a first dispensing of the following second-generation DAAs during the DAA period were identified: sofosbuvir (SOF), simeprevir (SMV), daclatasvir (DCV), sofosbuvir/ledipasvir (SOF/LDV), ombitasvir/paritaprevir/ritonavir (OBV/PTVr), dasabuvir (DSV), elbasvir/grazoprevir (EBR/GZR), sofosbuvir/velpatasvir (SOF/VEL), glecaprevir/pibrentasvir (GLE/PIB), sofosbuvir/velpatasvir/voxilaprevir (SOF/VEL/VOX). Initiation was defined as the dispensing of DAAs in the absence of any DAA delivery in the previous 6 months. DAAs were identified using a list of medicines based on the Anatomical Therapeutic Chemical classification.

### 2.3. Outcomes

During the DAA period, the following outcomes were studied: (i) annual and quarterly number of people tested for anti-HCV and patients initiating DAAs; (ii) annual distribution of the medical specialty of private sector prescribers of HCV tests; (iii) quarterly proportion of patients with DAAs dispensed in retail pharmacies and the annual distribution of the type of DAA prescribers (public or private sector and medical specialty only for physicians in the private sector); and (iv) demographic characteristics of people tested for anti-HCV and those initiating DAAs by 2-year periods.

For the COVID period, the observed monthly numbers of people tested for anti-HCV and patients initiating DAAs were presented. The expected monthly numbers of people tested for anti-HCV were also reported.

### 2.4. Data Analysis

We calculated the expected monthly and annual numbers of people tested for anti-HCV in 2020 and 2021 in the absence of the COVID-19 pandemic. For 2020 and 2021, these were estimated by applying the average trend coefficient observed between 2017 and 2019 to the observed numbers of people tested in 2019 and to the expected numbers of people tested in 2020, respectively. Then, we estimated the number of people not tested because of the COVID-19 pandemic by the difference between the expected and observed numbers of people tested in 2020 and 2021. 

Statistical analysis was performed using SAS Enterprise Guide software, version 7.15.

## 3. Results

### 3.1. Trends in HCV Testing and Treatment in the DAA Period 

#### 3.1.1. HCV Testing

The number of people tested for anti-HCV increased by 35.6%, from 2.678 million in 2014 to 3.633 million in 2021, with an increasing trend observed each year, except for 2020 when the number of screened people dropped by 8.3% (Figure 1). The annual percentage increase was higher in 2018 (+8.5%) and 2019 (+8.8%) compared to previous years (+3.3% in 2015, +5.8% in 2016, and +4.9% in 2017). In 2021, the observed number of tested people increased by 9.3% compared to 2020, but this was close to the number observed in 2019 (*n* = 3.622 million).

Private sector prescribers of anti-HCV tests were mainly general practitioners (Table 1). The proportion of general practitioners prescribing the tests increased from 44.0% to 51.9% for men and from 33.6% to 38.8% for women between 2014 and 2021. For women, obstetrician-gynecologists and medical gynecologists prescribed a substantial proportion of anti-HCV tests: 23.8% and 18.9% in 2014 and 2021, respectively. 

The age and gender distributions of people tested for anti-HCV remained stable, with a predominance of women (61.7–62.9% over 2-year periods) (Figure 2a). More than two-fifths of tested people were women under 39 years.

#### 3.1.2. DAA Initiation

For the DAA period, 96,776 DAA initiations were observed. The annual number of patients initiating DAAs increased sharply during the first half of the DAA period, from 11,500 in 2014 to 19,248 in 2017, before decreasing in the second half to 6972 in 2020 (−35.0% compared to 2019) and below 6000 in 2021 (Figure 1). 

More specifically, the quarterly number of patients initiating DAAs increased substantially from the second quarter (Q) of 2014 (Q2-2014), exceeding 3000 (Figure 3). Marked increases were observed in Q1-2015, concomitant with the DAA priority access policy and the availability of SOF/LDV, and in Q1 and Q2-2017 (with more than 5000 patients per quarter) when the HAS recommended universal access and EBR/GZR and SOF/VEL obtained market authorization. Subsequently, the quarterly number of patients initiating DAAs decreased in three successive steps: from Q3-2017 to Q2-2018 (about 4000 on average), from Q3-2018 to Q1-2020 (almost 2800 on average), and from Q2-2020 to Q4-2021 (about 1500 on average). Every year, except in 2020, the number of patients initiating DAAs was the lowest in Q3. 

Following the authorization for retail pharmacies to dispense DAAs in March 2018, the proportion of patients initiating this therapy dispensed by retail pharmacies reached 47% in Q2-2018, then progressively increased to 88% in Q4-2021.

Regarding the sector and specialty of DAA prescribers, between 2018 and 2021, the proportion of public hospital practitioners decreased from 84% to 72%, whereas in the private sector, the proportion increased from 13% to 21% for hepatologist-gastroenterologists and from 2% to 5% for general practitioners (Table 2). 

The age and gender distributions of patients initiating DAAs changed during the DAA period, with the proportion of men decreasing in the middle of this period from 65.0% in 2014–2015 to 57.0% in 2016–2017 and 59.0% in 2018–2019, before increasing in 2020–2021 (63.8%) (Figure 2b). The most represented age groups were 50–59-year-old men and women aged 60 years and over. Their proportion declined over the study period, while the proportion of men and women under 40 years and men aged 60 years and over increased.

### 3.2. Trends in HCV Testing and Treatment in the COVID Period

At the beginning of the COVID period during the first lockdown (17 March 2020–11 May 2020), the monthly number of people tested for anti-HCV and those initiating DAAs fell dramatically: −55.0% and −65.5%, respectively, in April 2020 compared to February 2020 (Figure 4). Then, the number of people tested went up more compared to the number of patients initiating DAAs, with the gap between these two curves continuing to increase. Decreases in the monthly number of people tested for anti-HCV and those initiating DAAs were more moderate during the second and third lockdowns (30 October 2020–15 December 2020 and 3 April 2021–3 May 2021, respectively). 

More specifically, compared to the same months in 2019, the numbers of people tested for anti-HCV decreased by 31.0%, 52.4%, and 25.6% in March, April, and May 2020, respectively (−36.2% overall) (Figure 5). In June 2020, the observed number of people tested exceeded the observed number in 2019 and even the expected number for 2020; it then followed the same trend and levels as in 2019 before decreasing in Q4-2020, just before and during the second lockdown. From December 2020, the observed numbers of tested people returned to at least the level observed in 2019 until a new decrease observed in April–May 2021 during the third lockdown; it then followed the same trend and levels as in 2019. 

Overall, it is estimated that 1.257 million people were not tested for anti-HCV due to the COVID-19 pandemic (613,000 and 644,000 in 2020 and 2021, respectively).

## 4. Discussion

Based on data from the SNDS, this descriptive study showed that 96,776 DAA initiations occurred between 2014 and 2021, with two main distinct phases: the first phase was marked by the rapid diffusion of DAA therapy from early 2014 and a load increase until mid-2017 following the successive implementation of new therapeutic strategies and policies regarding DAA access; the second phase was characterized by a gradual decrease in the number of patients initiating DAAs. This declining trend suggests that the reservoir of diagnosed and untreated patients is shrinking despite increased screening for HCV during the DAA period.

The upward trend in HCV testing has been observed in France since the early 2000s [14]. Data for the private sector since 2010 have shown a steady increase in the annual number of people tested for anti-HCV from 2010 to 2017, and more significantly from 2017 to 2019. This more marked surge in HCV testing between 2017 and 2019 may be partly due to the universal access to DAAs recommended since late 2016, as previously described in the United States [15]. This increase may also be explained by international and national hepatitis elimination targets [1,3], the authorization for rapid anti-HCV diagnosis tests in 2016, and multiple hepatitis screening campaigns in 2018 and 2019 [11,16]. 

Due to the COVID-19 pandemic, the number of people tested for anti-HCV dropped by 8.3% in 2020, with a substantial decrease during the restrictive lockdown of the first wave (−36.2% in March–May 2020 compared to March–May 2019, −52.4% in April 2020 compared to April 2019). The extent of this decrease was very close to the falls observed for HCV testing in other high-income countries such as Canada [17,18], United States [19,20], England [21], and Spain [22], or for HIV [23] and STI [24] testing in France. Despite a slight catch-up in June 2020, the monthly number of people tested for anti-HCV in 2020 and 2021 did not systematically return to the levels observed in 2019, notably because of the second and third lockdowns. We estimated that about 1.2 million people might not have been tested as a result of the COVID-19 pandemic. This estimate should be interpreted with care, as it is based on the growth rate observed between 2017 and 2019 and may thus be overestimated. However, the evolution rate in January and February 2020 was in line with that observed between 2017 and 2019. Conversely, the estimated number of people not tested because of the pandemic may be underestimated, because it did not take into account the biological tests performed in patients hospitalized in public hospitals. In 2021, the number of people tested was slightly higher than in 2019, suggesting that HCV testing has resumed, as confirmed by preliminary data from the first half of 2022.

In the private sector, unsurprisingly, anti-HCV tests were mainly prescribed by general practitioners (for more than half of men and nearly two out of five women in 2021). Almost 20% of tested women were prescribed the test in 2021 by gynecologists, mainly obstetricians. This result, along with the high proportion of women of childbearing age among the tested women, suggests that prenatal HCV screening is probably frequent despite not being recommended. The gender and age distributions of tested people remained constant during the DAA period, with a predominance of women and people under 40 years.

Our results confirm that access to DAA therapy rapidly expanded in France, particularly through compassionate use programs, with a large increase in the annual number of patients initiating treatment, which exceeded 10,000 until 2019 [25,26]. In the first half of the DAA period, the quarterly evolution in the number of people initiating DAAs was driven by both the availability of successive new therapies and the gradual broadening of access. The peak occurred in the first half of 2017 (with more than 5000 patients initiating DAAs per quarter), probably due to the availability of EBR/GZR and SOF/VEL (the latter being the first pangenotypic DAA) and the HAS recommendation for universal access to DAAs [7]. From this period onwards, the quarterly number of patients initiating DAAs dropped in three successive stages despite the availability of GLE/PIB (also pangenotypic) and several measures to facilitate DAA access, including the removal of the requirement for multidisciplinary hospital teams to assess non-complex cases, the extension of DAA dispensing to retail pharmacies, and the possibility for all physicians to prescribe DAAs for non-complex cases. This final measure has been only marginally adopted, as shown by the low proportion of general practitioners among prescribers in 2021; by contrast, dispensing in retail pharmacies was quickly and widely implemented. The age and gender distributions also evolved significantly over the study period. The proportion of men decreased during the mid-period before increasing, while the proportion of younger age groups for both genders and men aged 60 years and over increased. This trend reflects the different clinical profiles of patients able to gain access to DAAs following the successive implementation of priority measures as previously described [12,25,26].

The COVID period was marked by a dramatic decrease in the number of patients initiating DAAs in Q2-2020, particularly in April, followed by a substantial increase but without reaching pre-pandemic levels. The quarterly number then dropped again in the second half of 2021, being close to that observed during the first lockdown and less than in Q1-2014. The fall in Q2-2020 due to the restrictive lockdown was observed in similar proportions in other countries, as was the lack of a return to pre-pandemic levels [19,20,27]. For other drugs like HIV pre-exposure prophylaxis, initiations in France were impacted by the pandemic in 2020 but rapidly resumed an upward trend from 2021 onwards [28]. Although the COVID-19 pandemic evidently affected DAA initiations in Q2-2020, it is difficult to assess its impact on the evolution of DAA initiations for the remainder of the COVID period in light of the falling trend observed since mid-2017. This decreasing trend, also observed in Germany which shares similarities in terms of HCV epidemiology and public policy [29], suggests a depletion of the pool of diagnosed people requiring treatment. In 2016, it was estimated that 133,500 people (95% confidence interval (CI): 56,900–312,600) in the general population in metropolitan France had a chronic HCV infection, of whom only 107,600 (95%CI: 59,000–127,600) were diagnosed [30]. Our data included almost 97,000 DAA initiations during the DAA period, including 71,400 between 2016 and 2021. Compared to the estimated number of people diagnosed with a chronic HCV infection in the general population in metropolitan France in 2016 [30], this number corresponds to 66%, which is lower than the WHO’s 2020 target of 75% of people diagnosed with HCV starting antiviral treatment [1]. Considering the high efficacy of DAAs and the aging and mortality of HCV-infected individuals, the number of infected people (diagnosed and undiagnosed) has declined sharply in recent years [31]. Consequently, the identification of patients requiring treatment has become a challenging task that requires specific time-consuming interventions. In recent years, many interventions have been conducted in France with this goal, such as recontacting former patients not treated by DAAs and lost to follow-up by hepatology wards [32], or one-off screening interventions to enhance testing, linkage to care, and treatment initiations in the general population [33], at-risk populations [34,35], and in other countries [36]. In addition to screening and linkage to care, elimination efforts should also focus on infection and reinfection prevention measures. Currently, France is still considered to be on track to meet the WHO’s elimination targets by 2030 [37]. However, it is unlikely that the country reached the national mid-term objective of 120,000 people treated by the end of 2022 [3], as only 81% of this goal had been reached by late 2021.

The use of individual data from the SNDS, which covers almost the entire population in metropolitan France, was a major strength of this work. This database provides a comprehensive overview of the roll-out of DAAs as well as screening trends both before and since the availability of this treatment. The main limitations of this study were inherent to the medico-administrative nature of the SNDS such as the limited data on individual characteristics (no information on at-risk HCV exposure, fibrosis stage, disease severity, etc.). More specifically, the absence of information on key populations prevented identification of the high-risk populations affected by changes in tests and treatment initiations during the study period, and the highlighting of the populations that were particularly affected by the decrease in HCV testing and DAA uptake during the COVID-19 period. Furthermore, the screening information did not include tests performed in public in-patient care, sexual health centers (CeGIDD), or private laboratories without a medical prescription or among people without health insurance, which might represent about 14% of the total screening activity in metropolitan France in 2021 [38]. As only anti-HCV tests were considered for people tested for HCV, and not HCV RNA tests (recommended for many indications other than screening), it cannot be excluded that we missed a few previously infected and cured patients who were newly screened using only HCV RNA testing. Finally, the absence of these test results did not allow us to monitor the evolution of diagnosed people or the positivity rate among tested people. However, a repeated cross-sectional survey conducted among French laboratories highlighted a slight decrease in the anti-HCV positivity rate between 2016 and 2021 (from 0.73% to 0.67%) [38]. Given the high number of people treated during this period, the positivity rate of HCV RNA tests (not available in this survey) has probably decreased more sharply.

## 5. Conclusions

This nationwide observational study highlights the impact of successive new therapeutic strategies and access policies for DAA therapy on the load increase in DAA initiations until mid-2017. Since then, despite new measures to facilitate access to DAAs, the number of patients initiating treatment has sharply declined. This downward trend is indicative of the shrinking pool of people requiring treatment. Nevertheless, HCV testing activity increased during the DAA period despite the negative impact of the COVID-19 pandemic, which continued beyond the lockdowns. Implementing innovative and effective interventions to improve prevention, diagnosis, linkage to care, and treatment initiation, particularly among HCV high-risk and hard-to-reach populations (such as drug users, homeless people, and incarcerated people), is the new challenge to achieve HCV elimination in France.

## Figures and Tables

**Figure 1 viruses-16-00792-f001:**
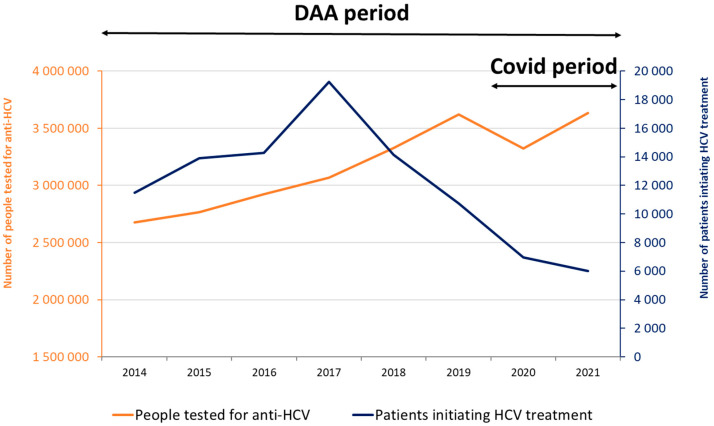
Annual number of people tested for anti-HCV antibodies and patients initiating HCV treatment, metropolitan France, 2014–2021.

**Figure 2 viruses-16-00792-f002:**
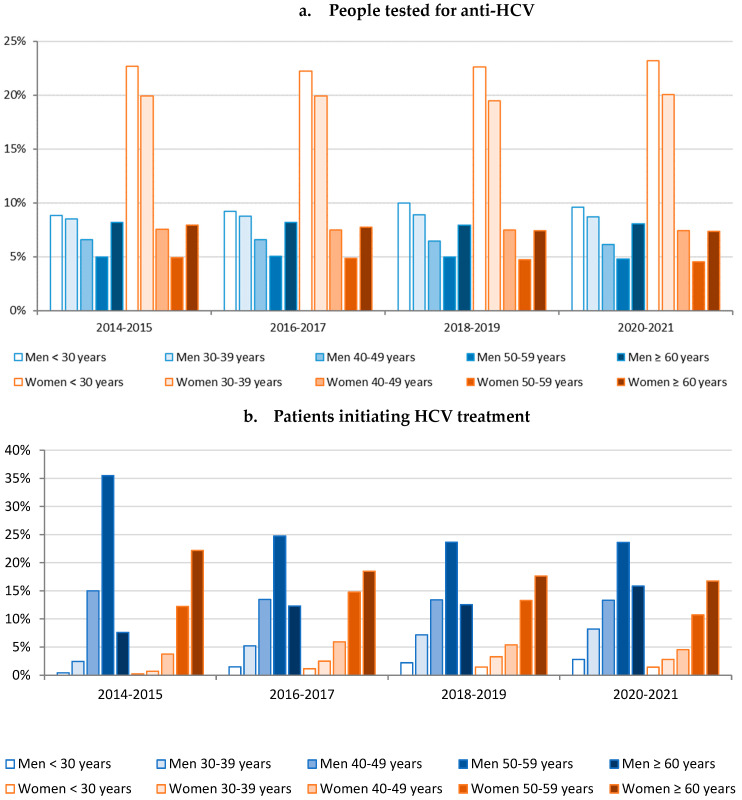
Age and gender distribution of: (**a**) people tested for anti-HCV; (**b**) patients initiating HCV treatment by 2-year periods, metropolitan France, 2014–2021.

**Figure 3 viruses-16-00792-f003:**
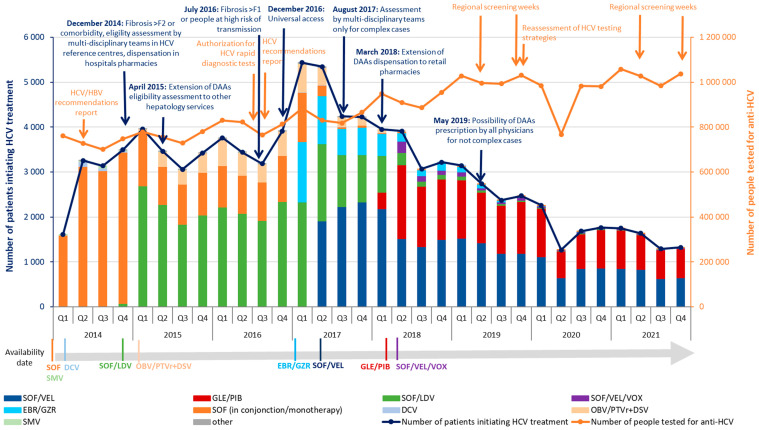
Quarterly number of people tested for anti-HCV antibodies and patients initiating HCV treatment by therapeutic strategy, metropolitan France, 2014–2021. SOF = sofosbuvir, SMV = simeprevir, DCV = daclatasvir, SOF/LDV = sofosbuvir/ledipasvir, OBV/PTVr =ombitasvir/paritaprevir/ritonavir, DSV = dasabuvir, EBR/GZR = elbasvir/grazoprevir, SOF/VEL = sofosbuvir/velpatasvir, GLE/PIB = glecaprevir/pibrentasvir, SOF/VEL/VOX = sofosbuvir/velpatasvir/voxilaprevir.

**Figure 4 viruses-16-00792-f004:**
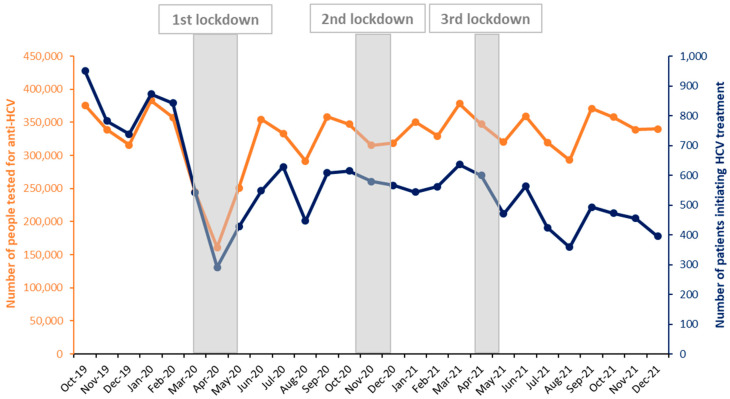
Monthly numbers of people tested for anti-HCV and patients initiating HCV treatment before and during the COVID-19 pandemic, metropolitan France, October 2019–December 2021.

**Figure 5 viruses-16-00792-f005:**
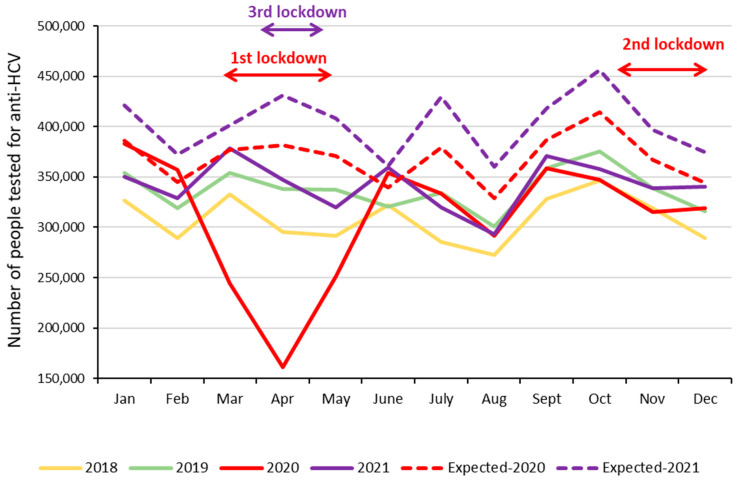
Monthly numbers of people tested for anti-HCV before and during the COVID-19 pandemic, metropolitan France, 2018–2021.

**Table 1 viruses-16-00792-t001:** Distribution of the medical specialty of private sector prescribers of reimbursed HCV tests, by gender of the person tested, in metropolitan France in 2014, 2018, and 2021.

Gender of the Person Tested		2014	2018	2021
Men	Number of reimbursements of HCV tests prescribed in the private sector	985,463	1,267,553	1,408,717
	Prescriber specialty			
	General medicine	44.0%	47.9%	51.9%
	Obstetrics-gynecology	3.9%	3.3%	3.5%
	Nephrology	2.9%	2.6%	2.6%
	Hepatology-gastroenterology	2.3%	2.1%	2.3%
	Orthopedic surgery and trauma	2.3%	1.7%	1.5%
	Cardiovascular pathology	2.1%	1.6%	1.5%
	Dermatology and venereology	1.2%	1.0%	1.1%
	Anesthesiology-surgical intensive care	3.2%	1.8%	1.2%
	Other specialties	7.0%	6.1%	7.3%
	Not specified	31.1%	32.0%	27.1%
	Total	100%	100%	100%
Women	Number of reimbursements of HCV tests prescribed in the private sector	1,698,614	2,086,228	2,358,207
	Prescriber specialty			
	General medicine	33.6%	37.8%	38.8%
	Obstetrics-gynecology	17.2%	15.2%	15.2%
	Medical gynecology	5.3%	3.9%	2.8%
	Rheumatology	0.8%	1.0%	1.0%
	Hepatology-gastroenterology	1.4%	1.3%	1.3%
	Orthopedic surgery and trauma	1.6%	1.3%	1.1%
	Nephrology	1.1%	1.0%	1.0%
	Obstetric and medical gynecology	1.3%	1.1%	0.9%
	Anesthesiology-surgical intensive care	2.3%	1.3%	0.8%
	Other specialties	4.9%	4.1%	4.5%
	Not specified	30.5%	32.0%	32.4%
	Total	100%	100%	100%

**Table 2 viruses-16-00792-t002:** Annual distribution of the sector and medical specialty of DAA prescribers, metropolitan France, 2018–2021.

Year	Public Hospital Practitioners *		Hepatologist-Gastroenterologists Private Sector		General Practitioners- Private Sector		Other Medical Specialties- Private Sector	
	*n*	%	*n*	%	*n*	%	*n*	%
2018	11,878	84%	1838	13%	283	2%	141	1%
2019	8043	75%	1930	18%	536	5%	214	2%
2020	4980	72%	1418	20%	449	6%	125	2%
2021	4308	72%	1236	21%	321	5%	132	2%

* For practitioners working only in the public sector, their specialty is not provided in the SNDS.

## Data Availability

The data that support the findings of this study were extracted by the authors from the French National Health Data System (SNDS). SNDS access is restricted to health insurance schemes and to a limited number of French public partners.

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
