# Peer review of "Impact of Public Policy and COVID-19 Pandemic on Hepatitis C Testing and Treatment in France, 2014–2021"

_viruses, 2024, doi:10.3390/v16050792_

Round 1

Reviewer 1 Report

Comments and Suggestions for Authors

Thanks for the opportunity to review this article. I have some comments which need to be addressed.

  1. The study lacks information on key factors like at-risk HCV exposure, disease severity, and fibrosis stage. it limits the ability of this study to identify the high-risk population affected by changes in tests and treatment uptake. How do you explain it? Is this the limitation of the study?
  2. why the study only included tests conducted within medical prescription excluding tests in public patient care, private laboratories, and sexual health centers. (Explain the reason).
  3. specific recommendations for effective interventions for future research and policy action are needed.

Author Response

The authors thank you for your insight and helpful comments. We have provided a detailed response to the comments below and have incorporated the suggestions in a revised and, we believe, improved manuscript.

  1. As the study is based on a medico-administrative database that is used for billing and reimbursed of health care, some useful epidemiological and clinical data, such as risk factors, results or biological or histological tests, are not available. This is the main limitation of the study, as already indicated. Clarifications have been made both in the methods and in the discussion sections.
  2. As added in the methods section, we used data from the French National Health Data system, that is a database used for billing and reimbursement of healthcare consumption. Therefore, tests performed without medical prescription, that are not reimbursed, are not included, as well as tests performed during a hospitalization in a public hospital or in a sexual health centers because they are not reimbursed individually (but supported by a global budget allocated to these structures). However, the comparison between the number of tests included in our study in 2021 (4,1 million) and the total number of tests performed in metropolitan France in 2021 (4,8 million) estimated by a survey conducted in all laboratories these tests (Brouard, BEH 2023), suggest that the tests not included in our study may represent about 14%. This information has been added in the discussion.
  3. The conclusion has been slightly expanded. However, in the absence of data on HCV risk factors and cascade of care data in this study, it seems difficult to go any further in terms of recommendations.

We hope that these revisions will be seen favourably for publication in Viruses.

Reviewer 2 Report

Comments and Suggestions for Authors

This is a well written paper and a thorough and thoughtful analysis on HCV testing and treatment among the general population in France; the authors pull from a data source that includes the majority of individuals in the country that received testing and the primary and secondary analysis that are highlighted make sense given the focus on pre- and post-COVID19 testing and treatment patterns.  A few specific comments are given below:

Abstract:

Abstract: The authors note from the data that “The declining DAA initiations since mid-2017 despite new measures improving access and screening efforts points to the shrinking pool of patients requiring treatment”; it would be more appropriate to say the declining DAA initiations could be due to a shrinking pool of patients requiring treatment, or a need to increase awareness among those who may have HCV infection but continue to not receive testing for HCV (i.e. those that may not be in touch with the healthcare system in France).

Main text:

Table 1: Is this correct that approximately 3-4% of HCV tests per year among men were performed by Obstetrics-gynecology specialty?

Figure 1 seems to indicate that there was a significant decrease in DAA treatment after a peak in 2017; given that the data came from administrative sources, why did the authors not have information on the number of people who tested HCV positive? The authors note that “declining trend suggests that the reservoir of diagnosed and untreated patients is shrinking despite increased screening for HCV during the DAA period.” Is certainly reasonable but the argument would be strengthened if there was more information provided on HCV prevalence/incidence among those that were tested during these periods. Although the authors do note this as a limitation in the paper, it would be beneficial if the authors could expand further as to why this information was not available for the study given the other rich data on tests and treatment regimens that were provided over the time period of analysis.

Do authors have any thoughts on the disparity in percentages with respect to HCV tests among men vs women? Would the majority of this difference be explained by the focus on testing pregnant women (even though the authors note that this is not a recommendation in France)?

It was interesting that 3/2018 with extension of DAAs to retail pharmacies, and 5/2019 DAA prescription by all physicians for not complex cases, there was a decrease in HCV treatment initiation after these points which is contradictory to what one would expect after these new guidelines were released, and it did not appear that testing for HCV had trended downward during these periods.  This could perhaps indicate that HCV positive tests had decreased during these periods which would indicate a decreased need for HCV treatment initiation (but again because there is no HCV positive test information, it is hard to know for sure if this can be explained by testing positivity).

Author Response

The authors thank you for your insight and helpful comments. We have provided a detailed response to the comments below and have incorporated the suggestions in a revised and, we believe, improved manuscript.

Abstract: Thank you for your proposal that has been added.

Main text:

  • Table 1: Indeed, a low proportion of men has been tested within a prescription by Obstetrics-gynecology speciality. The screening of the sexual partner may be proposed by gynaecologist-obstetrician on a case-by-case basis (e.g. for a woman with HCV exposures or newly diagnosed) or in the setting of specific prevention programmes. An example of such programme can be found at the following link: https://beh.santepubliquefrance.fr/beh/2024/8/2024_8_1.html (article in french with an abstract in English).
  • Figure 1: Unfortunately, the results of laboratory tests are not available in the database used in this study because the primary objective of this medico-administrative database is the billing and reimbursement of healthcare costs. These clarifications have been made both in the methods and in the discussion sections. Regarding the positivity rate of people tested, a repeated cross-sectional survey among French laboratories highlighted a slight decrease of the anti-HCV positivity rate between 2016 and 2021 (from 0.73% to 0.67%) (Brouard, BEH 2023, https://beh.santepubliquefrance.fr/beh/2023/15-16/pdf/2023_15-16_1.pdf ). Given the high number of people treated during this period, the positivity rate of HCV RNA tests (not available in this survey) has probably decreased more sharply. This information has been added in the discussion.
  • Regarding the disparity between men and women concerning HCV test:
  • Several elements suggest that HCV screening is probably largely performed during the pregnancy: i) the numbers of HCV tests performed and the sex and age distribution are similar to those for HBV ; ii) unpublished data on HBV prenatal screening show that HCV testing is often carried out simultaneously.
  • In general, the use of healthcare is higher among women than in men. For instance, French data from 2019 European Health Interview Survey (EHIS) showed that 88% of women reported at least one consultation with a general practitioner within the last year, compared with 80% of men (https://www.insee.fr/fr/statistiques/6047751?sommaire=6047805 )
  • Regarding the decrease in DAAs initiations despite new measures to enhance access to DAAs: we have few data, but feedback from professionals in the fields is valuable. According to hepatologists in university hospitals, the large majority of their patients HCV-infected were treated between 2014 and 2018 and newly referred patients for HCV infection become rare since then. This might explain the limited impact of these new measures. Currently, the critical points are access to testing and to care for HCV high-risk and hard to reach populations despite numerous innovative interventions. For instance, a major barrier to access to DAAs for PWID or homeless people is the lack of health insurance coverage. Research is being carried out in France on this topic.

We hope that these revisions will be seen favourably for publication in Viruses.